# Diagnosis of Chest Pneumonia with X-ray Images Based on Graph Reasoning

**DOI:** 10.3390/diagnostics13122125

**Published:** 2023-06-20

**Authors:** Cheng Wang, Chang Xu, Yulai Zhang, Peng Lu

**Affiliations:** 1School of Information and Electronic Engineering, Zhejiang University of Science and Technology, Hangzhou 310023, China; wcheng.zust@foxmail.com (C.W.); zhangyulai@zust.edu.cn (Y.Z.); 2Institute of Computer Innovation Technology, Zhejiang University, Hangzhou 310023, China; lupeng@zjuici.com

**Keywords:** deep learning, image classification, pneumonia diagnosis

## Abstract

Pneumonia is an acute respiratory infection that affects the lungs. It is the single largest infectious disease that kills children worldwide. According to a 2019 World Health Organization survey, pneumonia caused 740,180 deaths in children under 5 years of age, accounting for 14% of all deaths in children under 5 years of age but 22% of all deaths in children aged 1 to 5 years. This shows that early recognition of pneumonia in children is particularly important. In this study, we propose a pneumonia binary classification model for chest X-ray image recognition based on a deep learning approach. We extract features using a traditional convolutional network framework to obtain features containing rich semantic information. The adjacency matrix is also constructed to represent the degree of relevance of each region in the image. In the final part of the model, we use graph inference to complete the global modeling to help classify pneumonia disease. A total of 6189 children’s X-ray films containing 3319 normal cases and 2870 pneumonia cases were used in the experiment. In total, 20% was selected as the test data set, and 11 common models were compared using 4 evaluation metrics, of which the accuracy rate reached 89.1% and the F1-score reached 90%, achieving the optimum.

## 1. Introduction

Pneumonia refers to the inflammation of the terminal trachea, alveoli, and pulmonary interstitium. With the improvement of social medical care, improvement of nutritional status, popularization of related vaccines, and widespread use of antibiotics, the incidence and fatality rate of pneumonia have decreased to a certain extent; however, it still cannot be controlled at the ideal level. In particular, the impact of pneumonia on the life and health of the elderly and children [1] cannot be ignored. Studies have shown that pneumonia is still the leading cause of death among children from the neonatal period to the age of 5, and among the elderly over the age of 65 [2].

Imaging examinations, such as X-ray and CT, are important auxiliary examination methods for the diagnosis of respiratory diseases. According to the image display, the anatomical distribution of lesions in patients with pneumonia can be basically clarified, and thus classified into lobar pneumonia, lobular pneumonia, and interstitial pneumonia. Imaging results are also one of the important references for the timing of using of antibiotic and the selection of treatment plans [3]. Although the widespread use of CT has improved the early detection rate of patients with severe clinical manifestations, X-ray is still the first choice for clinical screening and imaging follow-up due to its advantages of simplicity, speed, and low cost. It is also the most commonly used examination method for chest diseases at present. The appearance of pneumonia on chest radiographs includes the thickening and blurring of early lung markings [4] and decreased transparency of the lungs. Typically, patchy shadows can be seen. In the current global pandemic of the new crown pneumonia epidemic, the demand for chest X-ray examination in hospitals has greatly increased, which requires radiologists to increase their ability to identify and distinguish pneumonia signs.

With the development of information technology and computer technology, image digitization and automatic recognition are widely used in various fields of society, which promote the transformation of production and lifestyle. Digitized images that are obtained by sampling, quantization, and encoding are scanned, identified, compared, analyzed by the computer, and information contained in the image is obtained, which can help radiologists make quick judgments under the premise of ensuring diagnostic efficiency. Compared with more complex computer technology application scenarios, such as face recognition [5] and iris recognition [6], the main difficulties affecting the interpretation of signs of chest radiograph pneumonia include the reflection of medical professional knowledge, such as the identification of similar signs, the identification of small lesions, the identification of areas near the mediastinum, and the identification of non-inflammatory small lesions nodules or old foci in the image information [7]. With the development of artificial intelligence, machine learning, and big data computing technology, the automatic image recognition of artificial intelligence has the potential to be applied. In the field of medical imaging, compared with the manual analysis by clinicians or radiologists, the recognition images made by computer technology can be automatically optimized by adjusting the contrast and so on [8], reducing the difficulty of image recognition. This article is based on a large amount of real clinical data, and quickly and accurately interprets the signs of pneumonia on the chest X-ray through the learning and training of artificial intelligence.

## 2. Related Work

In recent years, deep learning has developed rapidly and has been applied in many fields. Currently, intelligent driving [9], facial recognition [10], intelligent AI [11], and so on are all developed based on deep learning. At the moment of the epidemic, there is a shortage of medical resources, and most of the medical staff fighting on the front line cannot meet the current medical needs, which has prompted a large number of people to delve into the idea of intelligent diagnosis, and use artificial intelligence technology to solve the current problems that difficult to seek medical treatment and slow waiting.

Early on, machine learning was first applied in the medical field. Machine learning is often used to classify data (such as benign or malignant tumors [12]) or predict long-term systemic responses (such as wound healing time [13]). With continuous development, various machine learning algorithms suitable for disease diagnosis and prediction have been derived, including Alzheimer’s disease, cardiovascular disease, characteristic dermatitis, and even multi-type models for predicting up to 39 diseases. For example, Kim et al. [14] proposed a tree-based interpretable learning method to explore the optimal exercise therapy for patients with knee osteoarthritis, which generated interpretable processing rules by using random forests, which yielded low bias estimation but reduces the black-box aspects of machine learning algorithms.

However, deep learning is more suitable for image data than traditional machine learning. In machine learning, features for most applications need to be identified by experts and then hand coded according to the data type. The deep learning algorithm learns more advanced features from the data, which is a very unique part of deep learning and a part different from traditional machine learning. Deep learning reduces the task of developing a new feature extraction for each problem and tries to learn low-level features, such as edges and lines at an early level, such as convolutional neural networks (CNN), and then parts of the face. The last is the high-level representation of the face. Compared with machine learning, this method has a higher improvement in training time and cost. In view of the advantages of deep learning, a large number of scholars have applied it to the classification of pneumonia. Liang et al. [15] proposed a pneumonia binary classification model consisting of 49 layers of convolution-based on the idea of residual, using transfer learning to overcome the problem of insufficient data volume. For the data, about 5856 patients aged 1 to 5 in the Guangzhou Women and Children’s medical center, they finally achieved a good result on both the training and testing data. Wang et al. [16] used the new crown pneumonia data set named COVIDx and proposed the COVID-Net model that considers the diversity of the network structure. Experiments have proved that the model is better than VGG-19 and ResNet50. Shaban et al. [17] introduced a new hybrid diagnostic strategy to rank selected features by projecting them into the proposed patient space, constructing a feature connectivity graph (FCG), showing each the weight of a feature and the degree of combination with other features. Ozturk et al. [18] proposed a model for the automatic detection of COVID-19, which uses raw chest X-ray images and is called DarkCovidNet. The model aims to provide accurate judgments for binary classification (COVID-19 patients vs. normal) and 3-category (COVID-19 patients vs. normal pneumonia patients vs normal). The model made improvements based on DarkNet-19. Compared with the original DarkNet-19, DarkCovidNet uses fewer layers and filters, greatly improving performance. Li et al. [19] developed a unique voting algorithm that can accurately classify images into four categories (normal, bacterial pneumonia, viral pneumonia, and new coronary pneumonia). He combined 17 CNNs as a whole to generate an AI model to optimize the adaptability of the data, and the final result adopted a decision-making method of the majority rule. Bhandari et al. [20] proposed a lightweight convolutional neural network for the detection of new crown pneumonia, pneumonia, tuberculosis, and normal.

In this study, we propose an effective correlation reasoning network (graph-model), which is different from previous models. It is able to perform good region reasoning before the output of deep learning frameworks to achieve higher performance for image classification. The structure of this article is as follows. The third part mainly describes the structure of the proposed model, the fourth part introduces the experimental steps and results in detail, and the fifth part is a summary of the paper.

## 3. Method

In this section, we first detail the motivation and overall structure of the proposed method. Next, we detail the modules: adjacency matrix encoder and relation reasoning. Finally, we introduce a supervision strategy for training.

### 3.1. Motivation

For medical images about pneumonia, it is very difficult to automatically identify objects with high similarity to the surrounding environment. For general CNN-based deep learning frameworks, extracting features from images with extremely high intrinsic similarity is difficult to achieve accurate classification. The use of computers to discriminate diseases has gradually attracted the attention of many researchers. Inspired by the research on the visual system in biology, an ideal vision algorithm can achieve the reasoning and interaction of contextual information to a great extent in a given pneumonia medical image, so as to mine valuable clues. It is similar to the idea of identifying pneumonia in humans: from the perspective of the global image, establish the global dependencies between pixels, and make judgments based on the modeling of the dependencies to find the unusual. Intuitively, our target can benefit from mining valuable information in long-range dependency modeling and correlation reasoning. On the one hand, most of the existing medical image classification methods are implemented by the simple feature extraction of convolutional networks or modification of some convolution operations, and finally output the number of classifications by the full connection. On the other hand, they use additional convolutions to increase the depth or the width of the network to improve the recognition performance. However, these operations limit the exploration of the image area, and the correlation between the location information is not fully modeled and utilized. At the same time, the blind pursuit of the network dimension will bring less and less return on performance improvement. Exploring well-designed regional reasoning is beneficial for the classification of pneumonia before the logical output of the deep learning framework.

According to the above elaboration, we propose an efficient correlation reasoning network in this work to achieve higher-performance image classification. We first use the traditional convolutional network framework to extract features containing rich semantic information that is helpful for classification. Then we integrate the outputs of the last layer and the second layer of the baseline to generate an adjacency matrix, which expresses the degree of correlation of each region of the image. Finally, we use graph reasoning to complete the global modeling to help classify pneumonia diseases.

### 3.2. Overview

Figure 1 shows the overall design structure of our method. The proposed network is a two-stage encoding structure, including feature extraction, feature embedding, and the generation of the adjacency matrix. Considering that the information of the first layer of the baseline contains too much noise and insufficient effective information, for the generation of the adjacency matrix, we not only use the high-level semantic cues of the last layer but also integrate the rich spatial detail information contained in the second layer, which makes the embodiment of the correlation relationship more comprehensive and accurate. In our method, the generated adjacency matrix and embedded feature nodes are fed into the correlation reasoning, which can explicitly model long-range dependencies, providing guiding information in the process of learning classification. Finally, the feature information of reasoning is converted into logical output and supervised. Next, we briefly describe our two-stage structure.

In the first stage, given an input lung image $I\in R^{3\times H\times W}$, where *W* and *H* denote the height and width of the image, a traditional backbone network is adopted to extract multi-level features from five convolutional blocks, which are denoted as $\left\{C_{i}{|}_{i}=1,2,3,4,5\right\}$, where *i* indicates the convolutional block. The backbone we use here is MobileNet [21], and we discard the original fully connected logic output. For the realization of the following graph reasoning, we complete graph feature projection and adjacency matrix generation after the feature extraction of the backbone.

In the second stage, after we have feature projection $G_{s}\in C\times HW$ and adjacency matrix $\tilde{A}\in HW\times HW$, we feed them into the method of graph reasoning to model global dependencies and explore valuable information. The overall flow of the algorithm can be described as follows:(1)$$C_{i}=\mathrm{gnet}\left(I\right),i=1,2,3,4,5$$
(2)$$\tilde{A}=GAM\left(C_{5},C_{2}\right)$$
(3)$$out=FC\left(\mathrm{graph}\left(\tilde{A},G_{s}\right)\right)$$
where gnet represents the backbone called GoogLeNet. GAM represents the generation of the adjacency matrix in the first stage. graph denotes the operation of graph reasoning in the second stage. FC represents our final logical output operation, which is stacked fully connected operations. out is our final output.

### 3.3. The Generation of Adjacency Matrix

In this subsection, we elaborate on how matrix $\tilde{A}$ is defined to indicate the degree of correlation between feature regions. On the one hand, the adjacency matrix is aimed at the global correlation of features. Convolution only aggregates context in small local neighborhoods. Although redundant global calculations are naturally avoided, it is difficult to model global dependencies due to the limited receptive field; thus, directly using the feature information of the convolution output to generate the correlation will lead to the wrong reasoning of the information. On the other hand, the integration of spatial detail information will help to optimize the correlation, which is beneficial to generate a more accurate adjacency matrix. In addition, similar to most methods, we use the Euclidean distance to represent the degree of correlation between pixel *i* and pixel *j* in the adjacency matrix $\tilde{A}\in HW\times HW$. In this work, we implement it as the dot product and matrix multiplication.

Based on what was mentioned above, we divide the generation of the adjacency matrix into two steps as shown in Figure 1. In the first step, in order to assist CNN and benefit from the achievements of visual transformers in image processing in recent years, we embed a transformer block with multi-head self-attention after the final output of the convolution backbone, and perform subspace division on the current vector for attention calculation, realizing the attention calculation of the subspace, and then combine the calculation results with concatenation. This block naturally associates long-distance targets by comparing the global similarity. The process is as follows:(4)$$T=C_{5}+MHSA\left(norm\left(C_{5}\right)\right)$$
(5)M=T+mlp(norm(T))
where norm denotes the operation of LayerNorm. MHSA represents the multi-head self-attention. mlp denotes multilayer perceptron for forward propagation. At this step, we can obtain the optimized output *M*.

In the second step, we focus on the operation of the dot product and multiplication of matrices to obtain correlation coefficients representing Euclidean distances. We divided it into two branches as shown in Figure 2, which is called the feature encoder. In the first branch, we first use convolution with a 1×1 kernel to scale the features into a single-channel space to improve the computational efficiency. Then the spatial correlation is obtained by matrix transposition and matrix multiplication. We can obtain $A_{s}\in HW\times HW$ by the first branch:(6)$$A_{s}={Conv}_{1}\left(M\right)\times {Conv}_{1}{\left(M\right)}^{T}$$
where Conv1 refers to convolution with a 1×1 kernel. *T* is the transpose of the matrix, and × represent matrix multiplication.

In the second branch, for the input of the fifth layer, we use convolution to scale it to two channels, which corresponds to the number of our logical outputs, and for the input of the second layer, considering that the shallow features contain more noise, we first adopt the spatial attention unit to filter information from it, modeling the importance of spatial locations, finding the most important parts of the cues, and focuses on task-relevant regions. The feature information from the two layers is then combined by the dot product of the matrix and the maximum value method is used to obtain the single-channel feature of the combined information and the high-level information after dimensionality reduction. The degree of correlation is represented by the maximum probability value, which further enhances the reliability of the association. After the two information streams are extracted again by 1×1 convolution, respectively, matrix multiplication is used to obtain $A_{i}\in HW\times HW$:(7)$$h_{1}={Conv}_{1}\left(M\right)$$
(8)$$h_{2}=reshape\left(SA\left(C_{2}\right)\right)\cdot h_{1}$$
(9)$$A_{i}={Conv}_{1}\left(Max\left(h_{1}\right)\right)\times {Conv}_{1}\left(Max\left(h_{2}\right)\right)$$
where · represents the operation of the dot product, Max denotes the selection of the maximum value, SA represents the spatial attention, and reshape is an operation representing dimension matching.

Finally, the output results of the two branches are integrated by an element-wise addition to obtain the final correlation matrix $\tilde{A}$:(10)$$\tilde{A}=Sof\left(A_{s}\right)+Sof\left(A_{i}\right)$$
where + denote element-wise addition, Sof denotes the softmax function.

### 3.4. Graph Reasoning

Graph reasoning aims at mining and interacting with structured image data in the form of graph structure. We need to convert the three-dimensional image data into a two-dimensional matrix that depends on the graph structure. We first project the feature space of the image. For the feature nodes of the graph, here, each pixel of the feature map is a node, which is a C×N matrix, where N(HW) is the total number of pixels, that is, the number of graph nodes, and *C* is the feature dimension of each node. Given an input feature map S∈C×H×W, we project it to $G_{s}\in C\times HW$ so that we construct a graph space containing HW nodes, in which each node has the feature dimension of C×1.

After obtaining feature node $G_{s}$ and adjacency matrix $\tilde{A}$, we feed them into graph convolution. Our goal is to explore valuable information by modeling long-range dependence and obtain $G_{out}\in C\times H\times W$ after graph reasoning, which is defined as
(11)$$G_{out}=\sigma \left(\tilde{A}G_{s}W\right)$$
where σ denotes the ReLu activation, and *W* is a trainable parameter. Here, we use a one-dimensional convolution with a convolution kernel of 1×1. In order to preserve the original information and enhance the robustness of the features, we reuse the output of the graph convolution layer with the input
(12)$$G_{out}=G_{out}+G_{s}$$

Finally, we reconstruct the output $G_{out}\in C\times HW$ on the graph space to C×H×W. As described above, we feed the output of graph reasoning into stack of fully connected layers to obtain the logical output of the classification:(13)$$out=FC\left(G_{out}\right)$$

### 3.5. Supervision Strategy

The proposed method obtains the logical output of b×2 at the end. We optimize the training parameters by supervising this output and computing their cross-entropy loss as follows:(14)$$l=-\sum_{x=1}^{2}G\left(x\right)\log\left(S\left(x\right)\right)+\left(1-G\left(x\right)\right)\log\left(1-S\left(x\right)\right)$$
among them, G(x) and S(x) represent the ground truth and predicted value of the corresponding category, respectively. We aim to judge whether there is pneumonia disease, so the logical output category is 2.

## 4. Experiment and Results

In this section, we first introduce the details of the data set we used in this paper, then introduce the parameters in the model processing, experimental configuration, and model evaluation methods, and finally state the experimental results of the model.

### 4.1. Data

The data set contains 6189 pediatric pneumonia diagnostic images, including 3319 normal cases and 2870 pneumonia cases. In deep learning, the bigger the number of data we have, the better the scores we obtain. Sufficient data can not only improve the training accuracy of the model but also prevent overfitting of the model in the case of complex models. Moreover, sufficient data can better simulate various cases in real scenarios, making the model more expressive. To this end, the data-enhancement method is used to process the original data of the existing data set in the following angles to enhance the expression of the original data: the horizontal and vertical rotation of the image, the rotation of the random angle, the change of the image brightness, chroma, contrast and color temperature, and the addition of some random noise. Considering that the size of each image is different and there are too many redundant features, in order to unify the model input, the image is scaled to the pixel size of 480 × 480, and the range of pixel values is normalized to the interval of 0∼1. In addition, considering the redundancy and interference of irrelevant features in the non-target area of the image, a center cropping operation with an aspect ratio of 0.9 was adopted for all images, that is, the shaded parts on both sides and the parts other than the upper and lower heads and ribs were removed as shown in Figure 2. The reason for this is that, from a physician’s perspective, most pneumonia judgments only consider the lungs, and the role and contribution of the areas outside the lungs to the diagnosis are not very large, which can better reflect the clinical significance of the operation. The expanded data are shown in Table 1. There were 5974 normal cases and 5166 pneumonia cases, with a total of 11,140 samples.

Figure 3 shows the comparison of the image data before and after enhancement. From the human perspective, this is just a simple image flip, but for the computer, these are two different pictures. Because the computer, after reading the picture, is the size of each pixel in the picture, each pixel value is different to show different information, so it is much more sufficient and more accurate than the information observed by human eyes. The value of each pixel in the picture shown in Figure 2 is different. Therefore, for the computer, the data-enhancement operation can expand the number of samples in a favorable way, increase the performance ability of the data, and simulate a variety of complex situations in the real scene as much as possible so that the model has stronger generalization ability and can give more accurate prediction in other situations.

### 4.2. Experimental Setup

We divide the data set into a training set and test set according to the ratio of 0.8. The number of samples in the training set is 9902, the number of samples in the test set is 1238, and the number of model iterations is set to 150 epochs. We adopt a minibatch stochastic gradient descent with a momentum of 0.9 to train our mode. The weight decay and initial learning rate are set to 0.0005, and the batch size of the experiment is set to 16. We optimize the parameters of the model through continuous iteration on the training set and, finally, evaluate the performance of the model through the test set. Since this is a classification task, in order to better evaluate the classification accuracy and performance of the model, we use four metrics based on the confusion matrix—accuracy, precision, recall, and F1-score—to evaluate the overall performance of the model.

### 4.3. Results

As shown in Figure 4, the result of the model about the classification of pneumonia on the test set, the confusion matrix is based on true positive (TP), true negative (TN), false positive (FP) and false negative (FN), which is a common method to evaluate the performance of the model. TP represents the number of samples that were correctly identified as positive by the model, TN is the number of samples that were correctly identified as negative by the model, FP is the number of samples that were misclassified as negative by the model, and FN is the number of samples that were incorrectly classified as positive (pneumonia) by the model. The upper left and lower right corners of the confusion matrix represent the percentage of true positives and true negatives in the total sample, while the lower left and upper right corners represent the percentage of false positives and false negatives in the total sample, respectively. It can be found that the number of true positives and true negatives correctly predicted by the model is more than the number of false positives and false negatives, while the number of incorrect predictions is very small. The F1-score index can also reflect the superiority of the model from another perspective.

In order to reflect the superiority of the model in this task, Figure 5 lists the comparison results of 12 models in the pneumonia classification task, in which 4 metrics are, namely, the accuracy, precision, recall, and F1-score, respectively.

Metrics of vgg11, vgg13, and vgg16 models shown in Figure 5 indicate that when using the vgg framework, the deeper the network, the better the improvement of the accuracy, recall, and F1-score. Similarly, for resnet18, resnet34, and resnet50, the deeper the network, the better the metrics with the accuracy and precision. For the accuracy and recall metrics of Densenet121 and DenseNet161, although the deeper the network structure, the higher the accuracy. However, the deeper the network, the larger the number of model parameters and the longer the inference time.For example, the accuracy of Densenet121 to Densenet161 is only improved by 0.2%. The enhancement of the network has a limited impact on the metrics, and when designing the framework, we must consider not only the number of parameters in the model, but also the enhancement of the effect due to the number of parameters.Considering the limitations of the number of model parameters in practical applications, we compare different backbone models with a small number of parameters, including Vgg11, ResNet18, Densenet121, Inception v3, Convnext, and Mobile v3. Mobile v3 is a prominent model in lightweight networks, and it can be found that there is not a big gap between the metrics of this network and other models through comparison. Therefore, Mobile v3 is used as the backbone to explore this basis. Based on the adjacency matrix and graph reasoning modules, the model in this paper reaches the optimal method.

It can be found that for accuracy, our method achieved an improvement compared with the best model, Densenet161, and reached the optimal performance of 0.891. Similarly, for the precision, it surpassed the Densenet121 model and reached 0.888, and for the F1-score, our model surpassed the Densenet121 and reached 0.900. The above three metrics reached the optimal performance in the participating comparison model.

Figure 6 shows the accuracy of the model on the training set and the test set. In the experiment, we set 150 epochs. Red represents the change curve of the accuracy rate during the training process, and green represents the change curve of the accuracy rate during the testing process. It can be found that as the number of iterations increases, the accuracy of the model on the testing set is almost stable at the 50th epoch, but the accuracy on the training set is still improving. Therefore, it can be considered that subsequent training will not improve the generalization ability of the model. Based on this, the optimal selection of weight is retained in the 50th epoch. To further analyze the performance of our model, we visualize the parameters of the model weight using Grad-CAM [22].

Figure 7 is the heat map after the visualization of Grad-CAM. It can be seen that the trained model can pay more attention to the double lung area, which is the red area in the heat map. It indicates that our model accurately uses effective features to make decisions. This is also in line with the judgment criteria of clinicians.

## 5. Discussion

In this section, we systematically analyze the proposed method in comparison with other algorithms based on the experimental results of the previous section and discuss the advantages of the algorithm as well as the current problems.

### 5.1. Comparison with Related Algorithms

Common classification algorithms for machine learning include logistic regression, K-nearest neighbor, SVM, and parsimonious Bayes. However, these classification algorithms have some limitations when used for image classification. First, the dimensionality of image data is very high, which can cause some distance or density-based algorithms (e.g., K-nearest neighbor and parsimonious Bayes) to be very time consuming and space consuming in terms of computation and storage. In addition, image data contain a large amount of noise, distortion, illumination, scale, etc., which can affect the feature extraction and recognition of image data, resulting in some linear or global feature-based algorithms (e.g., SVM), failing to effectively capture the local and nonlinear features of the image data. Second, there may be significant differences or similarities between different classes of medical images, which may lead to the inability of some hypothesis-based or a priori algorithms to accurately depict the distribution and probability of image data. For these reasons, machine learning algorithms are not applicable to this data set.

As shown in Figure 5, deep learning has developed rapidly in the last two years, and this paper uses the common classification models in recent years as a comparison. The advantages and disadvantages of these models are summarized as follows:**Vgg** [23]: The advantage of vgg is its simple structure and easy implementation of algorithms that use multiple repeated convolutional and pooling layers to build deep networks. However, it is relatively computationally intensive, takes a long time to train, and its fully connected layers account for a large proportion of the parameters relative to the overall model.**ResNet** [24]: ResNet uses the idea of residuals to solve the problem of difficult training of deep networks with strong generalization ability, but its network structure is more complex and requires more computational resources.**DenseNet** [25]: This is an algorithm that uses dense connectivity to enhance feature propagation and feature diversity. Its advantage is that it can effectively utilize low-level features, reduce gradient disappearance and overfitting problems, and improve classification performance; its disadvantage is that it requires more memory space and computational resources.**Inception** [26]: This is an algorithm that uses multi-scale and multi-branch convolutional structures to extract features at different levels and degrees of abstraction. Its advantage is that it can adapt to images of different sizes and shapes to improve classification accuracy and efficiency; the disadvantage is that the network structure is more complex and requires more parameter tuning.

From the above, it can be seen that the deep learning algorithms that have achieved good results in the past two years often have a huge number of parameters, which especially occupy memory and computational resources. Therefore, in this paper, we adopt the lightweight framework mobileNet as the backbone and use deep separable convolution to reduce the number of parameters and computation, which can improve the running speed and efficiency, and combine with the graph convolution model structure [27]. Graph convolution can take advantage of the lightweight and efficiency of mobileNet, as well as its own adaptability and local perception ability to handle the classification of data and improve the accuracy of the model.

### 5.2. Disadvantage

In this paper, we combine the graph convolution algorithm to improve the accuracy of mobile networks, but other problems arise. First, graph convolution occupies a large amount of memory space when constructing the adjacency matrix, which is not conducive to storage and computation. Second, the nonzero elements of the adjacency matrix reflect the direct connection relationship between nodes, and when considering the influence between nodes with larger distances, multiple multiplication operations are required for the adjacency matrix, which will increase the computational complexity and sparsity.

## 6. Conclusions

In this paper, we applied deep learning to solve medical imaging problems, make improvements to medical imaging and achieve good performance. We proposed a model with relevant reasoning based on the feature extraction of the backbone using the mobileNet network. We first used the low-level spatial detail information and high-level rich semantic information to generate an adjacency matrix by matrix point multiplication and matrix multiplication operations, which represent the degree of correlation of each pixel in the feature matrix. Then, inspired by the idea of graph convolution, our method establishes long-distance relationships and model global dependencies from a global perspective. Finally, through modeling and reasoning on dependencies, we can focus on the inflammatory lesion area to judge the occurrence of pneumonia. We compared 11 classic models including the backbone we used, and adopted 4 evaluation indicators, in which 3 indicators achieved the best results: the accuracy reached 0.891, the precision reached 0.888, and the f1-score reached 0.900. This shows that our proposed algorithmic model is superior to using only the baseline model.

In future work, we will increase the diversity of data to improve the difficulty of model classification. Medical data sets are difficult to collect, with label inconsistency and noise interference, including subjective factors of physicians, so new data augmentation and self-supervised learning methods need to be developed to improve the data utilization and representation capability of models. In addition, medical research often encounters scenarios where diagnoses cannot be made with a single piece of data and usually needs to rely on multiple examination data to identify diseases. Therefore, it is necessary to combine image data with other related tasks to achieve multimodal data input in order to improve the generalization capability of the model.

## Figures and Tables

**Figure 1 diagnostics-13-02125-f001:**
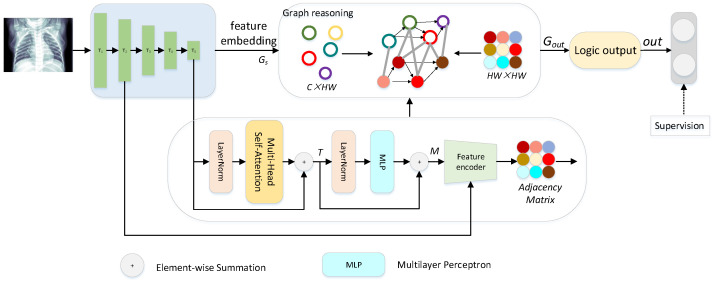
Overall architecture of proposed network.

**Figure 2 diagnostics-13-02125-f002:**
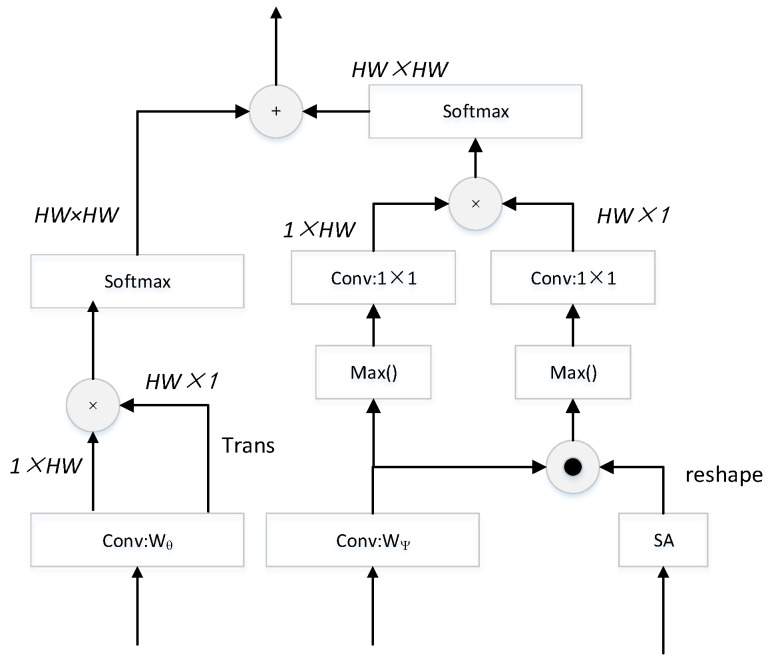
Feature encoder.

**Figure 3 diagnostics-13-02125-f003:**

The original image is on the (**left**), and the rotated image is on the (**right**) after the data-augmentation operation.

**Figure 4 diagnostics-13-02125-f004:**
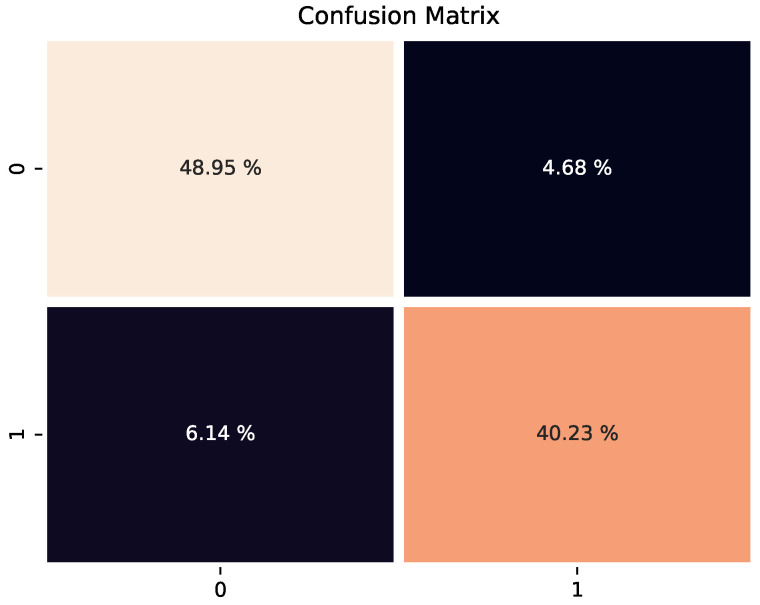
Confusion matrix.

**Figure 5 diagnostics-13-02125-f005:**
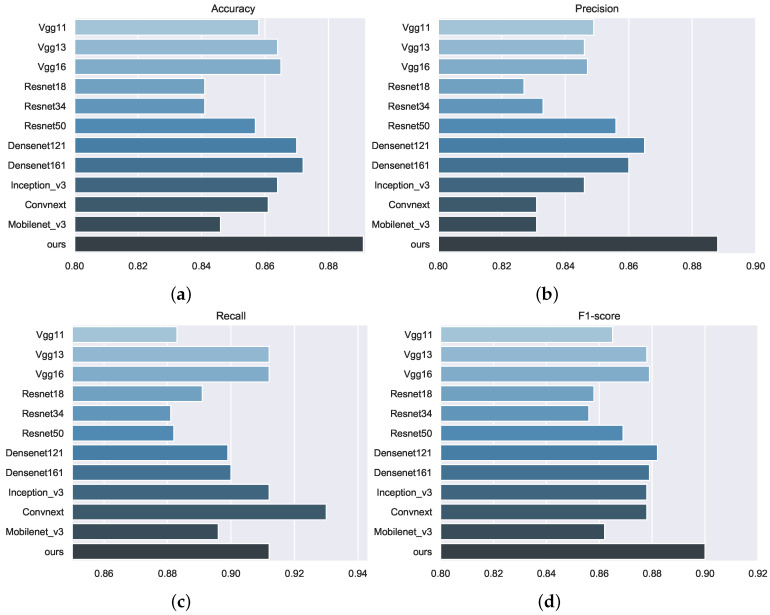
It is a comparison of the model metrics, (**a**) is Accuracy, (**b**) is Precision, (**c**) is Recall, and (**d**) is F1-score.

**Figure 6 diagnostics-13-02125-f006:**
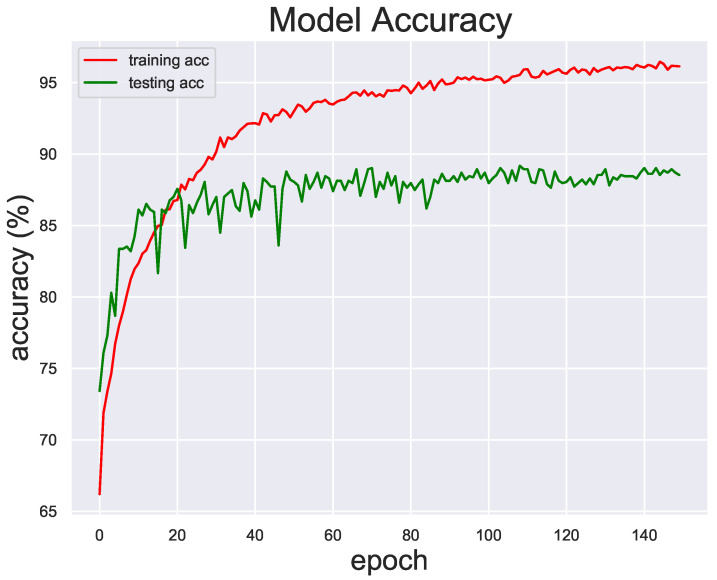
Training and test accuracy curves.

**Figure 7 diagnostics-13-02125-f007:**
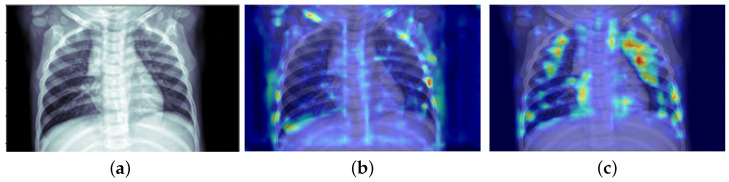
(**a**) is the original image, (**b**) is the heat map generated by the comparison model, and (**c**) is the heat map generated by our model.

**Table 1 diagnostics-13-02125-t001:** Data set.

Images	Stage1	Stage2
Normal	3319	5974
pneumonia	2870	5166
Total	6189	11,140

## Data Availability

Not applicable.

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
