# Peer review of "Diagnosis of Chest Pneumonia with X-ray Images Based on Graph Reasoning"

_diagnostics, 2023, doi:10.3390/diagnostics13122125_

Round 1
Reviewer 1 Report
1- Designing of the paper is not appropriate.
2- English language and spelling need to improve.
3- The authors should update the discussion section with more information on the different methods of chest imaging.
4- The manuscript has potential but needs thorough revision. Emphasis should be given to explain about of more recently work.
5- In the final section, the author should emphasize on future work and the limitation of the present work.
English language ans grammar spelling need to improve.
Reviewer 2 Report
The paper focuses on classifying chest pneumonia X-ray images using machine learning techniques. Although the authors have utilized algorithms well with decent accuracy, I have a few concerns mentioned as follows.
-
I do not see any novelty in the paper. If the author checks google scholar or other relevant search engines, many papers have been published using the same dataset and machine learning has been performed for classification.
-
Other relevant papers have also attained decent accuracies such as up to 90% or more detection rate.
-
Using existing datasets is fine, however, a novelty in some form has to be there in order to get published. Using the machine learning classification technique alone is not a novelty especially when in the past a lot of work has been done on it which happened particularly during COVID time on this dataset.
-
I recommend rejection based on no novelty in the paper. I would highly recommend authors introduce some novelty in the paper and re-submit.
NA
Reviewer 3 Report
I congratulate the authors for the excellent experimental article produced
Round 2
Reviewer 2 Report
The authors have clearly addressed all my comments. Thanks for that!
I believe the manuscript is suitable for publication now. I just have some minor comments that are provided as follows.
1. Figure 4, confusion matrix - I would suggest changing the digits of the confusion matrix to percentages. In that way, it is easy for the reader to observe the results and also shows professionalism. For instance, change the 606, 58, 76, and 498 to the form of percentages. This can be done by a simple function in Python.
2. Present the results in Table 2 also in a bar chart graph so it is easy for the reader to compare the results. It can also be done using the Python matplotlib library.
Once the above changes are performed, I would recommend acceptance of the manuscript.
Please improve the quality of English by eliminating minor grammatical errors. This can be done by using platforms like Grammarly or some other relevant tools.
